# Mast Cells and Basophils in IgE-Independent Anaphylaxis

**DOI:** 10.3390/ijms241612802

**Published:** 2023-08-15

**Authors:** Krzysztof Pałgan

**Affiliations:** Department of Allergology, Clinical Immunology and Internal Diseases, Collegium Medicum in Bydgoszcz, Nicolaus Copernicus University, Ujejskiego 75, 85-168 Bydgoszcz, Poland; palgank@wp.pl

**Keywords:** mast cell, basophil, anaphylaxis, IgE, IgG, anaphylatoxins, MRGPRX2, complement

## Abstract

Anaphylaxis is a life-threatening or even fatal systemic hypersensitivity reaction. The incidence of anaphylaxis has risen at an alarming rate in the past decades in the majority of countries. Generally, the most common causes of severe or fatal anaphylaxis are medication, foods and *Hymenoptera* venoms. Anaphylactic reactions are characterized by the activation of mast cells and basophils and the release of mediators. These cells express a variety of receptors that enable them to respond to a wide range of stimulants. Most studies of anaphylaxis focus on IgE-dependent reactions. The mast cell has long been regarded as the main effector cell involved in IgE-mediated anaphylaxis. This paper reviews IgE-independent anaphylaxis, with special emphasis on mast cells, basophils, anaphylactic mediators, risk factors, triggers, and management.

## 1. Introduction

Anaphylaxis is a severe systemic allergic reaction with a range of clinical manifestations. The “International Consensus on Anaphylaxis (ICON)” described anaphylaxis as “a serious, generalized or systemic, allergic or hypersensitivity reaction that can be life-threatening or fatal” [1]. According to the European Academy of Allergy and Clinical Immunology’s (EAACI), anaphylaxis is a life-threatening reaction characterized by acute onset of symptoms involving different organ systems and requiring immediate medical intervention [2]. 

Epidemiological data indicate that the prevalence of anaphylaxis has increased worldwide over the last two decades [3,4]. The prevalence of anaphylaxis in Europe is estimated to be 0.3% (95% CI, 0.1–0.5), and in the United States of America, 1.6% to 5.1% [4,5,6,7]. The increasing prevalence of anaphylaxis is caused by food, nonsteroidal anti-inflammatory drugs, insect stings, monoclonal antibodies, and chemotherapeutic agents [8,9]. 

In anaphylaxis, cutaneous and mucosal symptoms (urticaria, erythema, angioedema, pruritus) occur most frequently—in 80–90% of cases. Cardiovascular symptoms characterized by an acute increase in vascular permeability, relaxation of vascular smooth muscle, constriction of vessels in the thoracic cavity, palpitations, tachycardia, and hypotension, are present in >50% of cases [10]. Interestingly, cardiovascular and obstructive lung diseases increase the risk of severe anaphylaxis. Respiratory symptoms include bronchoconstriction, dyspnea, cough and wheezing. Less common, however, is the involvement of gastrointestinal symptoms such as emesis, diarrhea, conditions of the nervous system (confusion, drowsiness, seizure disorders) [11,12]. However, up to 10% of patients with anaphylaxis can have no cutaneous or mucosal manifestations [11,13]. 

## 2. Triggers of Anaphylaxis

The most frequent causes of anaphylaxis are foods (peanut, hazelnut, milk, egg, wheat, celery, shellfish, peach) [14,15,16], venoms of hymenoptera (wasps, bees, fire ants) [17,18], drugs (antibiotics, particularly beta-lactams, non-steroidal anti-inflammatory drugs), and radiocontrast media (high-osmolality ionic contrasts, low-osmolality non-ionic contrasts) [19,20,21,22]. Numerous studies have now demonstrated that foods are the most common elicitors of anaphylaxis in children and young adults. Epidemiological studies have estimated that fatal food-induced anaphylaxis rates range from 0.03 to 0.3 deaths per million inhabitants per year [23,24,25,26,27]. 

## 3. Co-Factors and Risk Factors of Anaphylaxis

The available data from studies describe several risk factors for severe anaphylaxis and fatal outcomes. The highest rates of fatal outcomes of drug-related anaphylaxis occur at an older age, between 55 to 85 years. Increased drug exposure and cardiovascular diseases may be related to increased prevalence of severe anaphylaxis [28,29]. 

A retrospective study reported that asthma and IgE sensitization predict an increased rate of anaphylaxis [30,31]. It has been reported that stress, infection, non-steroidal anti-inflammatory drugs, alcohol, older age, mastocytosis, and medications, such as beta-blockers or ACE inhibitors, may increase its severity [32,33]. 

A delay in adrenaline injection or incorrect administration have been associated with significantly higher odds of a severe reaction [34]. 

## 4. Mechanisms of Anaphylaxis

Anaphylactic reactions have a heterogeneous nature. Recently published reports on anaphylaxis indicate a broad range of effector cells are involved, including mast cells, basophils, dendritic cells, T-cells, neutrophils, macrophages, platelets, endothelial cells, smooth muscle cells, and natural killer cells (NK) [35,36]. 

Numerous studies have described the importance of mast cells and basophils in the anaphylactic reaction [37]. Mast cells were discovered by Paul Ehrlich in 1879 and are recognized for their versatile role in a range of immunological responses [8]. About one hundred years later, in 1970, the IgE crosslinking high affinity FcεRI receptor was described. Since then, studies have widely demonstrated the mechanism of mast cell activation during allergic response and anaphylaxis [38]. 

From a pathophysiological point of view, these cells can be activated by several triggers acting on receptors on the cell surface [9,39]. The most important mechanism underlying anaphylaxis is the IgE-mediated reaction involving mast cells and basophils. Crosslinking of the high-affinity IgE receptors on mast cells and basophils by IgE–antigen complexes lead to a massive release of biologically active mediators [40,41]. In this activation a stem cell factor (SCF) augments mast cells IgE-dependent mediator release [42].

Interestingly, recent human studies of anaphylaxis have suggested that basophils play a key role in food-mediated anaphylaxis [40]. The mediators released from activated mast cells and basophils, including histamine, platelet-activating factor (PAF), enzymes, cysteinyl leukotrienes (CysLTs), and anaphylatoxins, are responsible for the signs and symptoms of anaphylaxis [19,43]. Table 1 characterizes basophils and mast cells. 

The importance of IgE mediated reactions to common allergens are a well-established cause of anaphylaxis and was demonstrated 50 years ago. Much accumulating evidence and a number of studies support the belief that there exists an IgE-independent mechanism of anaphylaxis [58]. 

## 5. IgE-Independent Anaphylaxis

Certain factors directly cause a release of anaphylactic mediators from mast cells and basophils [59]. Table 2 summarizes the main differences between IgE-dependent and IgE-independent anaphylactic reactions. 

In 2006, Tatemoto et al. first demonstrated a receptor called Mas-related G protein-coupled receptor X2 (MRGPRX2) in mast cells [60]. MRGPRX2 is coupled with seven transmembrane G proteins, and recently it has been shown that it is expressed on basophils and eosinophils [61]. Activation of this receptor is responsible for non-IgE-mediated anaphylactic reactions. Based on these findings, it has been shown that MRGPRX2 plays an important role in the activation of mast cells and induction of anaphylactic reactions by neuromuscular blocking drugs (NMBDs; such as atracurium, mivacurium, tubocurarine, rocuronium, cisatracurium), opiates (codeine, morphine), antibiotics (vancomycin, ciprofloxacin, levofloxacin, moxifloxacin), iodinated contrast media, plasma expanders and dyes [62,63].
ijms-24-12802-t002_Table 2Table 2The most important differences between IgE-mediated and IgE-independent anaphylactic reactions [1,52,63,64,65,66,67,68,69,70,71,72,73,74,75].IgE-Dependent AnaphylaxisIgE-Independent AnaphylaxisDegranulation and release anaphylactic factors from mast cells and basophils by high affinity receptors FcεRIDegranulation and release of anaphylactic factors from mast cells and basophils by: -IgG antibodies binding to Fc gamma receptors (FcγRs),-MRGPRX2 receptor activated by drugs and biogenic amines and peptides,-Complement system,-Direct action.Triggers: allergens (food, venoms, drugs)Triggers: nonsteroidal anti-inflammatory drugs, opiates, contrast media, antibiotics, anesthetics.The risk (1% to 20%) of biphasic reactions, UnknownRequire a period of sensitization.May occur on first exposure to an agent (allergen). Idiopathic anaphylaxis


The most commonly reported adverse event of morphine is pruritus. Some reports suggest that opioid receptor agonists can engage a mast cell-independent mechanism to trigger pruritus [64].

## 6. Anaphylatoxins

The complement system is part of innate immunity and adaptive immunity [76]. Anaphylatoxins C3a and C5a are among the components of the system. These soluble, small (consisting of 77 and 74 amino acids, respectively), highly cationic proteins target a broad spectrum of immune and non-immune cells. C3a and C5a have receptors (C3aR, C5aR) on various cell types. Several reports have shown expression of C3aR and C5aR on human neutrophils, basophils, eosinophils, mast cells, monocytes/macrophages, dendritic cells (DC), T-cells and endothelial cells [65,66]. These anaphylatoxins bind to specific G protein-coupled receptors on mast cells and basophils and thereby stimulate their degranulation [77]. 

Numerous clinical observations support the importance of anaphylatoxin-mediated anaphylaxis [78,79]. It can be induced by a dialysis membrane, protamine neutralization of heparin, liposomal drug infusion, infusions of Cremophor EL, polysorbate 80 and polyethylene glycol by direct activation of the complement components [67,68,80]. 

Furthermore, the involvement of the complement system in the development of anaphylaxis is correlated with the severity of the reaction [81]. 

Activation of human MC_TC_ cells via the crosslinking of FcεRI results in the release of tryptase. In vitro studies have demonstrated that mast cell-derived β-tryptase generates C3a and C5a from C3 and C5, respectively [82,83]. In addition to the direct action of anaphylatoxins on mast cells and basophils, some reports suggest that a component of peanuts and wasp venom—mastorpan—acts synergistically with IgE-dependent mast cell activation [84]. These observations suggest that complement activation exacerbates anaphylaxis induced by other mechanisms [69]. 

## 7. IgG-Mediated Anaphylaxis

In humans, the existence of IgG-mediated anaphylaxis is controversial. Numerous reports document IgG anaphylaxis mainly in animal experimental models [70]. Experiments in mice have demonstrated the induction of anaphylaxis following intravascular antigen administration. IgG antibodies act through FcγRs on macrophages, neutrophils, dendritic, cells, platelets, basophils, and mast cells. Platelet-activating factor (PAF) was identified as the most important mediator in IgG-mediated anaphylaxis in immunized mice [71,72,85,86]. Human IgG-mediated anaphylaxis has been reported following the infusion of large quantities of dextran, aprotinin, and von Willebrand factor. On the other hand, numerous reports have described IgG-dependent anaphylaxis in patients who were treated with a biologic therapeutic (intravenous immunoglobulin, variety of chimeric, humanized and fully human monoclonal antibodies) [87,88,89,90]. 

To contrast the presented data, it is worth mentioning the report by Lessof et al. [91]. This researcher purified immunoglobulins from beekeepers, who naturally develop high serum concentrations of bee venom-specific polyclonal IgG antibodies due to frequent sting events, which mediated protection from sting challenge-provoked anaphylaxis in patients allergic to bee venom. Additionally, studies of a subcutaneous allergen immunotherapy have shown 10- to 100-fold increases in the serum concentrations of IgG. In particular IgG_4_ can compete with IgE for the allergen and prevent crosslinking of high-affinity IgE receptors (FcεRI) on basophils and mast cells, thereby inhibiting the release of histamine and other anaphylactic mediators. Repeated low-dose exposure to the allergen leads to limited inflammation [92,93]. The mast cells and basophils express inhibitory receptors—FcγRIIb, that suppress the IgE-mediated activation. Recent studies have shown that FcγRIIb takes part in the maintenance of peripheral tolerance. The FcγRIIb inhibits the maturation of dendritic cells, allergen presentation and T cell priming as well as decreases the release of β-hexosaminidase, histamine, LTC-4, MIP1-α, and TNF-α from mast cells and basophils, and decreases anaphylaxis. This receptor also inhibits macrophage phagocytosis mediated by FcγR [60,94].

## 8. Idiopathic Anaphylaxis

The term of idiopathic anaphylaxis or spontaneous anaphylaxis refers to diagnosed anaphylactic reactions when no cause can be identified [95]. According to recent publications a global incidence of this anaphylaxis ranged from 50 to 112 episodes per 100,000 person-years [96]. The clinical picture of idiopathic anaphylaxis is similar to other known causes of anaphylaxis. In idiopathic anaphylaxis mast cell activation leads to a measurable increase in tryptase levels during attacks, with a return to baseline afterwards [97]. Possible mechanisms underlying the pathophysiology of idiopathic anaphylaxis include hidden allergens in food, increased activation of T and B lymphocytes and basophils, mastocytosis, mast cell activation disorders, exercise-induced anaphylaxis and medication [73,98]. 

The reactions due to hidden food allergens represent a major health problem for sensitized persons. The anaphylaxis caused by hidden allergens can induce a wide variety of reactions [99]. Foods can contain many allergic proteins present in small amounts. It has been shown that common food allergens are celery, mustard, lupin, sumac, pectin, carmine, mycoprotein, and *Anisakis*. The literature has demonstrated the very wide range of foods involved in hidden anaphylaxis and it is not possible to include them all [100,101]. 

Mastocytosis is a heterogenous collection of disorders characterized by pathological proliferation and activation of atypical mast cells with variable organ system involvement [101]. Most studies on mastocytosis reveal the somatic c-kit D816V mutation and elevated baseline levels of serum tryptase [102]. 

The available epidemiological data about the exact prevalence of mastocytosis are limited. However, it is accepted that mastocytosis is a relatively rare, and the prevalence has been estimated in adults to be 10–17 per 100,000 inhabitants [103]. In mastocytosis, the mast cells are abnormal and in a hyperactive state, and degranulate spontaneously or in response to different kinds of triggers. For example, factors such as physical agents (heat, cold), drug and medications (acetylsalicylic acid, non-steroidal anti-inflammatory drugs, muscle relaxants, iodized contrasts, beta-adrenergic blockers, antagonists of cholinergic receptor), infections (viruses, bacteria), surgery or endoscopy, some hymenoptera venom components (mast cell degranulating peptide, mastorpan M) and dextran can lead to mast cell degranulation [104,105,106]. 

This condition has been shown to be associated with increased severity of allergic and anaphylactic reactions and may interact variably with primary and secondary mast cell disease, resulting in complex combined disorders. The prevalence of sting-induced anaphylaxis in adult patients with mastocytosis ranges from 20–30% [107]. The anaphylactic reactions in the majority of cases with systemic mastocytosis are characterized by the increased severity and predominance of cardiovascular symptoms [17,108].

Hereditary alpha-tryptasemia (HαT) is an autosomal dominant genetic trait characterized by elevated basal serum tryptase ≥8 ng/mL due to increased pro-alpha-tryptase synthesis rather than increased mast cell activation [109]. The incidence of HαT has been estimated at 5% in the general population [110]. The most commonly reported clinical symptoms among individuals with hereditary alpha tryptasemia are flushing, irritable bowel syndrome, urticaria, angioedema and anaphylaxis. Significant associations were observed between elevated basal serum tryptase and connective tissue abnormalities such as joint hypermobility, retained primary dentition, congenital abnormalities, orthostatic hypotension, tachycardia, presyncope and syncope, chronic pain and fatigue [111].

Basophils and mast cells contain preformed mediators (histamine, tryptase, chymase, carboxypeptidase, heparin and some cytokines) and those synthesized during and after cell activation (prostaglandin D2, leukotriene C4, PAF, cytokines: IL-4, IL-5, IL-6, transforming growth factor-β—TGF-β, and tumor necrosis factor α—TNF-α). Therefore, these mediators are divided into two groups: preformed granule products, released firstly and secondarily, or formed de novo by mast cells and basophils mediators [110]. Histamine and PAF are crucial mediators in the development of signs and symptoms of anaphylaxis. Their blood levels correlate with the severity of anaphylaxis [52,112,113,114].

## 9. Histamine

Histamine, a biogenic amine, was isolated from liver and lung tissue in 1927. The name histamine was given after the Greek word *histos* (’ΙΣΤOΣ)-tissue [115]. Histamine is synthesized in various cells throughout the body, including gastric mucosa parietal cells, mast cells, basophils, lymphocytes, central nervous system neurons and is stored primarily in the granules of mast cells and basophils [116]. Interestingly, recent studies have demonstrated that resident gut microbial species such as *Enterococcus faecalis*, *Bifidobacterium pseudocatenulatum*, Lactobacillus gasseri, Escherichia coli, Morganella morganii and *Proteus mirabillis* are able to produce histamine [117,118]. 

Histamine is present in a wide range of foods in highly variable concentrations. The accumulation of histamine in food is the result of the transformation of amino acids by gram-positive and gram-negative bacteria. Microbiologically altered food (fish, meat, cheeses) or fermented products (beer, red and white wine) are the main exogenous source of histamine [119]. 

The biogenic amine has a low-molecular-weight and is synthesized from l-histidine exclusively by histidine decarboxylase [120]. This enzymatic reaction was first described by Windaus and Vogt in 1907 [121]. 

The degradation of histamine was shown to be via diamine oxidase (DAO) and histamine methyl transferase [122]. DAO deficiency predisposes to histamine intolerance. Decreased activity diamine oxidase can be caused by certain pathologies (bowel pathologies, hemotherapy, gastrointestinal surgery) polymorphisms in genes encoding the enzyme or pharmacological factors such as chloroquine, clavulanic acid and verapamil [123]. 

Histamine plays an important role in physiologic functions (embryonic development, hematopoiesis, cell proliferation, regeneration, wound healing, gastric acid secretion and acts as a neurotransmitter) and pathology (allergic inflammation, anaphylaxis, hypotension, cardiac arrhythmias, bronchospasm, rhinitis, cramping, diarrhea, and cutaneous wheal and flare responses) [124,125]. Histamine G-protein-coupled receptors: H_1_, H_2_, H_3_, and H_4_ mediate the effects of histamine [126]. 

The H_1_ receptor is expressed in endothelial cells, airway cells, vascular smooth muscle cells, dendritic cells, monocytes, neutrophils, T cells, B cells, hepatocytes, chondrocytes and nerve cells [127]. Activation of H_1_ results in capillary dilation and enhanced vascular permeability and bronchoconstriction. In addition, H_1_ activation leads to synthesis of nitric oxide, arachidonic acid, prostacyclins, increased chemotaxis of eosinophils and neutrophils at the site of inflammation and promotion of IgE production [128,129]. 

The H_2_ receptor was identified pharmacologically in 1972. The H_2_ receptor is strongly expressed in the stomach. Clinically used H_2_ receptor antagonists, including cimetidine and ranitidine transformed and became the cornerstone in the treatment of gastric ulcers [130,131]. The antagonists of histamine H_2_-receptors inhibit acid secretion stimulated by histamine [132]. The existence of H_2_ was also confirmed on leucocytes, macrophages, mast cells, neutrophils, thrombocytes, erythrocytes, vascular smooth muscle cells, fibroblasts and cardiac cells [133,134]. The receptors inhibit the release of histamine on basophils and mast cells, mediate basophil suppression, the suppression of antibody synthesis, T cell proliferation, and cell-mediated cytolysis. This receptor might also contribute to early protective mechanisms during the buildup phase of venom immunotherapy (VIT) [135]. Histamine, through the H_2_-histamine receptors, induces cardiac arrhythmias [136,137]. 

The H_3_ receptor has been identified in the central and peripheral nervous system as a presynaptic receptor controlling the release of histamine and other neurotransmitters. This receptor is mainly involved in brain function and mast cells [138]. 

The histamine H_4_ receptor is the last discovered member of the family of histamine receptors. The H_4_ receptor is expressed in the bone marrow and peripheral blood hematopoietic cells (neutrophils, eosinophils, T cells), spleen, thymus, lung, small intestine, colon, testes, and kidney [139,140]. This receptor modulates mast cell degranulation indirectly by inducing upregulation of FcεRI and mediating histamine-induced eosinophil chemotaxis [141,142]. 

Alcohol intake is an augmenting risk factor of anaphylaxis [143]. One hypothesis suggests that alcohol intake is associated with increased total serum IgE levels. Another hypothesis suggests that drinking red wine and/or ethanol increases histamine levels by inhibiting the enzymes responsible for the degradation of histamine in the intestine, such as diamine oxidase. It is believed that ethanol may also increase absorption of food allergens causes the anaphylactic reaction [29]. Alcohol can also directly activate mast cells, and basophils and thus increase severity of the anaphylactic reaction [144].

## 10. Platelet Activating Factor

Platelet-activating factor (PAF) was described in 1974 by Jacques Benveniste [145]. In the literature, the term PAF generally refers to 1-alkyl-2-acetyl-*sn*-glycero-3-phosphocholine [146]. PAF is produced and released by neutrophils, eosinophils, mast cells, endothelial cells, fibroblasts and epithelial cells [147]. PAF binds to the PAF receptor on platelet, endothelial cells, airway smooth muscle, monocytes, macrophages, and neutrophils [148,149]. PAF is a potent anaphylactic mediator [150]. Interestingly, this factor can be rapidly released (about 30 s) upon mast cell and basophil stimulation. PAF level is an indicator of severity, presumably of IgE-mediated anaphylaxis, and plays the most important role in the pathogenesis of anaphylactic shock [122,151]. This factor induces hypotension by vascular dilatation, increases permeability, decreases coronary blood flow, and decreases myocardial function [152,153]. PAF leads to dyspnea, wheezing from bronchospasm, throat tightness, and odynophagia or hoarseness associated with laryngeal edema or oropharyngeal angioedema [52]. In a generalized anaphylactic reaction, the skin is involved first. The effect of PAF is most obvious in generalized pruritus, flushing, and urticaria. Apart from that, PAF activates neutrophils and eosinophils, and induces platelet aggregation [154].

The role of IgG in anaphylactic reactions has also been demonstrated in mouse models. In an animal model some studies have shown that PAF, rather than histamine, plays the most important role in anaphylaxis. Neutrophils and macrophages, under experimental conditions have been reported to be major PAF producers in IgG-mediated anaphylaxis [86,155].

The biological half-life of PAF, which ranges from 3 to 13 min and is determined primarily by the rate of inactivation by PAF-acetylhydrolase (PAF-AH). PAF-AH activity was inversely correlated with the severity of anaphylaxis [36,150]. 

The catalytic activity of PAF-AH is regulated by its association with low density lipoproteins (LDL), as lowering the LDL by statins increases the half-life of PAF [156]. There are several studies suggesting that PAF acetylhydrolase deficiency is an independent risk factor for fatal anaphylaxis, separate from other anaphylactic cofactors [157]. 

Furthermore, patients with the lowest levels of PAF-AH activity are 27 times more at risk of severe or fatal anaphylaxis than patients with normal levels of PAF-AH activity [114]. 

## 11. Leukotrienes

Leukotrienes (LT)s are also lipid, inflammatory, and anaphylactic reaction mediators synthesized de novo by arachidonic acid by the 5-lipooxygenase (5-LO) pathway. LTs are produced mainly by mast cells, basophils, eosinophils and macrophages in response to a variety of factors [56]. These molecules are able to increase vascular permeability, mucus hypersecretion, bronchoconstriction, leukocyte chemotaxis, and airway responsiveness [158]. Especially, leukotriene C_4_ (LTC_4_) has been extensively studied in the context of allergy and anaphylaxis. Genetic variation in the LTC_4_ synthetic pathway may explain the differences in clinical pictures of anaphylactic reactions [74,159]. LTC_4_ are synthesized following degranulation of mast cells, basophils and potentiated anaphylactic reactions by increasing the permeability of endothelium, enhancing vasodilatation, inducing bronchoconstriction, mucus hypersecretion, and recruiting other inflammatory cells. These molecules increase anaphylaxis duration and severity by stimulating leukocyte chemotaxis [58,160]. 

## 12. Mast Cell Proteases

Mast cell proteases are released from activated mast cells during degranulation. They have a wide variety of functions in allergic reactions, host defense and immune modulation [161].

## 13. Tryptase

The finding that tryptase is released from mast cells alongside other mediators, including histamine and other anaphylactic mediators, is useful for a correct diagnosis of anaphylaxis [162]. The half-life of tryptase is about 1.5–2.5 h. During anaphylaxis serum levels begin to rise about 5–30 min after the event, reach a peak after 1–3 h and return to the basal value within 16–24 h from the end of the event [34]. However, the World Allergy Organization anaphylaxis guidance recommends evaluating baseline serum tryptase at least 24 h after resolution of anaphylaxis symptoms, even when tryptase concentration during the episode remains within the normal range [53]. 

Apart from diagnostic aspects, tryptase is involved in pathogenesis anaphylactic reactions. For instance, tryptase increases the expression of CCL2 and the pro-inflammatory cytokines IL-6 and IL-22. It has been shown that tryptase through the activation of bradykinin stimulates vascular permeability [163]. In the late phase of anaphylactic reaction, tryptase increases neutrophil and eosinophil chemotaxis. It is also thought that tryptase can increase bronchoconstriction, stimulate smooth muscles of the bronchi and can interact with the complement and clotting cascades [164]. 

## 14. Chymase

Chymases belong to the large family of serine proteases. It has been reported that chymase expression in mast cells is constitutive and can account for up to 25% of the total cellular protein of these cells [165]. However, it was also demonstrated that chymase is a potential biomarker for anaphylaxis. Other studies have shown that blood chymase concentration is <3 ng/mL in cases without anaphylaxis and during anaphylaxis increases after 1 h up to 89.8 ng/mL and remains raised for 8–24 h [166,167]. Mast cell tryptases and chymases receive more attention, because they are almost entirely mast cell-specific, whereas other proteases, such as cathepsins G, C, and L are expressed by a variety of inflammatory cells [168]. 

The latest research indicates that the novel basophil granule protein, termed basogranulin, could be used as a suitable serum marker of anaphylaxis [169,170]. 

## 15. Treatment Aspects of IgE-Independent Anaphylaxis

Acute management of allergic or nonallergic anaphylaxis includes the early use of IM epinephrine. Epinephrine administration is widely accepted by major guidelines as first-line therapy [34,171]. Other drugs, such as antihistamines and corticosteroids, are not recommended for the initial treatment of anaphylaxis [172]. The latest research indicates that corticosteroids and antihistamines are not life-saving and do not prevent biphasic anaphylaxis [173]. Indeed, there is some evidence that corticosteroids may be of benefit in poorly controlled asthma and in cases of refractory anaphylaxis [174,175,176]. 

Prevention of mast cell activation, degranulation, and secretion of anaphylactic mediators is the ultimate goal. Interestingly, some flavonoids (luteolin, tetramethoxyluteolin, quercetin, genistein) inhibited mast cells and decreased histamine secretion in mast cells [40,54,55]. Syed et al. [177], showed that lactic acid inhibited mast cell degranulation by attenuating the MRGPRX2 receptor.

## 16. Conclusions

Anaphylaxis is a multiorgan reaction involving a broad range of effector cells and mediators. Classically, most cases of anaphylaxis are IgE-mediated. A variety of allergens can induce IgE synthesis. IgE-independent anaphylactic reactions are caused by factors independent of IgE antibodies. Mast cells and circulating basophils release anaphylactic mediators following the binding of an allergen, crosslinking the cell surface with the high-affinity IgE Fc receptors, crosslinking IgG with FcγR, activating the complement system, or activating the membrane MRGPRX2 receptor. Mixed reactions involving both IgE- and non-IgE-mediated pathways can occur as well. In the future, development of high-affinity MRGPRX2 inhibitors may provide useful therapies for drug-induced anaphylactic reactions. 

## Figures and Tables

**Table 1 ijms-24-12802-t001:** Main features of mast cells and basophils.

	Mast Cells	Basophils	References
Origin	Bone marrow. In mouse a phenotypic identification of mast cell progenitors (MCps) was made in fetal blood. The development of MCps into mast cells is dependent on stem cell factor (SCF). SCF is the most important factor in differentiation, proliferation, survival and function of mast cells and their progenitors. (IL-3) supports development and function.	Bone marrow. The research performed in mouse blood cell hematopoiesis suggest basophils develop from hematopoietic stem cells (HSCs) via myeloid progenitors (CMPs): granulocyte-monocyte progenitors (GMPs), and granulocyte progenitors (GPs) in the bone marrow. Further development continues in the bone marrow or in the spleen.	[44,45]
Location	Different tissue.	Peripheral blood.	[45,46,47]
Types	MC_TC_- store tryptases, chymases, and carboxypeptidases. Prevail in the skin, lymph nodes, lung and the gut submucosa. Express C5aR. MC_T_- contain only tryptases and prevail in the intestinal and pulmonary mucosa. Mast cells expressing tryptase and carboxipeptidase A3. Localized in the airway epithelium in asthmatic subjects and esophageal in patients with eosinophilic esophagitis. MC1- anti tumorigenic mast cells MC2- tumorigenic mast cells	IL-3-induced line of basophils. Thymic stromal lymphopoietin (TSLP)-induced line of basophils. Lower responsiveness to IgE/antigen complexes.	[48,49] [50,51] [38,40,41]
Function	Mast cells and basophils play a crucial role in anaphylaxis.		[45,46,52,53]
	Associated with infections to certain parasites (*Sarcoptes scabiei*, *Strongyloides ratti*, *S. brasiliensis*, *Schistosoma mansoni*, *Leishmania*, *Toxoplasma*, *Trypanosoma*, *Plasmodium*). Carboxypeptidase A, tryptase β, chymase play an important role in resistance to the venoms of hymenoptera and snakes. Mouse mast cell protease 4 stimulates immune responses to the venoms of the Gila monster lizard (*Heloderma suspectum*) and some species of toxic scorpions. Antimicrobial activity (against *S. Aureus by* ejecting extracellular DNA traps) may be activated by a variety of bacterial and viral products.	Take part in innate immunity.	[46] [47,50]
Life span	Months	Days	[54,55]
Tryptase content	High.	Low, about 500 times less than in mast cells.	[48,56,57]
Basogranulin	Unknown.	released upon both IgE-dependent and IgE-independent stimulation.	[52]

## Data Availability

Not applicable.

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
