# Peer review of "Mast Cells and Basophils in IgE-Independent Anaphylaxis"

_ijms, 2023, doi:10.3390/ijms241612802_

Round 1

Reviewer 1 Report

The review is interesting and written in good English. The subject matter is analyzed with great detail and completely as far as the author has proposed. I have no particular remarks to make and the manuscript can be accepted in its present form for publication, after some minor revisions.

1) in table 1 there is text in red, please correct.

2) on page 8 you went to a new line before the end of the sentence "gastrointestinal surgery) polymorphisms in genes encoding the en-zyme or pharmacological factors such as chloroquine...", please correct.

3) In Table 2 check the formatting of the sentence "Triggers: nonsteroidal anti-inflammatory drugs, opiates, media, antibiotics and anesthetics"; you went to the top before the end of the sentence, please correct.

4) references 159, 160 and 161, 162 and 188 are written in red, please correct.

5) On page 8 in the sentence "surgery or endoscopy, hymenoptera venoms and dextran can lead to mast cell degranulation [95, 96]" you could add a bibliographical note and cite the recent work of Bava et al "Therapeutic Use of Bee Venom and Potential Applications in Veterinary Medicine" Vet. Sci. 2023,10, 119. https://doi.org/10.3390/vetsci10020119 which explains which are the components that determine this mast cell degranylation.

Author Response

I would like to thank the reviewers for their kind evaluation of my article. All comments have been included in the corrected version. I hope that the current version of my work will be approved by the reviewers.

Reviewer 2 Report

Minor assumptions:

·        The biggest weakness of the manuscript are table 1 and 2. Maybe one could transform this table into a visual representation of the functions of mast cells and basophils. Further, the caption to the tables is too short. Please provide more information there.

·        There are a lot of extra spaces in the sentences. Please remove them

·        Last sentence of chapter 4: Try to find here a more elegant transition to the IgE independent mechanism. Maybe mention a little anecdote or similar.

·        Chapter 9: There in the first sentences “performed mediators” are mentioned. Explain this in more detail

Author Response

(The authors gave the same response as above.)
